# Low Self-Esteem and Life Satisfaction as a Significant Risk Factor for Eating Disorders among Adolescents

**DOI:** 10.3390/nu15071603

**Published:** 2023-03-25

**Authors:** Agnieszka Pelc, Monika Winiarska, Ewelina Polak-Szczybyło, Justyna Godula, Agnieszka Ewa Stępień

**Affiliations:** Department of Dietetics, Institute of Health Sciences, Medical College of Rzeszow University, University of Rzeszow, 35-959 Rzeszow, Poland; ap108063@stud.ur.edu.pl (A.P.);

**Keywords:** self-esteem, eating disorders, bulimia, losing weight, adolescent

## Abstract

Background: Eating disorders are a problem that is becoming more and more common among younger and younger age groups. Many studies examine the risk factors for EDs, however, the treatment of these diseases is very complicated and requires dietary, psychological and medical intervention. Methods: 233 primary and secondary school students aged 12 to 19 were surveyed using the EAT-26 (Eating Attitudes Test-26) questionnaire, the self-esteem Scale SES and the Cantril scale for life satisfaction. Results: Women, when compared to men, showed lower self-esteem, satisfaction with their appearance, body weight and their lives and at the same time a higher risk of eating disorders in all three areas. Low life satisfaction is often correlated with weight loss greater than 10 kg. Low self-esteem correlated positively with significant weight loss (>10 kg) and more frequent uncontrollable binge eating and exercising (more than 60 min a day) to influence appearance. People with low self-esteem were more likely to be treated for EDs. Subjects dissatisfied with their lives binged, feeling that they could not stop. Conclusion: The younger the person, the more likely they are to develop eating disorders. This is closely correlated with low self-esteem and negative life satisfaction. Men were more likely to be satisfied with their weight, appearance, and life, and were less likely to show ED symptoms.

## 1. Introduction

Eating disorders (ED) are a complex group of mental disorders that result in pathological eating behaviors that can lead to serious complications [1]. Incorrect attitudes towards body weight and shape and the foods eaten play a key role in the onset and maintenance of eating disorders [2]. The term describes a range of irregular eating behaviors that may or may not lead to a diagnosis of a particular disorder [3]. All eating disorders significantly impair physical health and interfere with psychosocial functioning [4]. Eating disorders can affect people of all ages, gender, sexual orientation, ethnicity and geographic origin. Teenagers and young adults are especially at risk [5]. Studies indicate an increase in the prevalence of eating disorders in recent years, from 3.5% in 2000–2006 to 7.8% in 2013–2018 [6]. Only 50% of patients respond to modern treatments, and about 20% develop permanent disorders [7]. Anorexia has the highest mortality rate: up to 20% of patients [8]. Therefore, a multidisciplinary approach from the medical community is necessary, i.e., a psychiatrist, a psychologist, and a dietitian. However, multidisciplinary collaboration in the treatment of eating disorders remains uncommon, in part due to high costs [9].

In the Diagnostic and Statistical Manual of Mental Disorders (DSM-4), eating disorders include anorexia nervosa (AN), bulimia nervosa (BN), and eating disorder not otherwise specified (EDNOS). EDNOS is a composite diagnosis that includes all eating disorders that do not meet the AN or BN diagnostic criteria [4]. The fifth edition of the Diagnostic and Statistical Manual of Mental Disorders (DSM-5) identifies the three eating disorders listed above, and additionally recognizes the importance of subthreshold and atypical states by naming five subtypes of other specified feeding and eating disorder (OSFED):Atypical anorexia nervosa (i.e., features of anorexia without low weight);Nervosa bulimia (of low frequency and/or limited duration);Binge eating disorder (of low frequency and/or limited duration);Purging disorders;Midnight Eating Syndrome.

The DSM-5 also includes a category called unspecified feeding or eating disorder (UFED), which includes people who do not fit into any of these five categories or for whom there is not enough information to make a specific diagnosis of OSFED [4]. In addition to changes in classification, the diagnostic criteria for all types of eating disorders have been changed in the DSM-5. The weight loss requirement was relaxed, and the amenorrhea requirement was removed in AN. According to the DSM-5, binge eating episodes should occur at least once a week for a period of three months [10].

Although the different forms of eating disorders differ in course and treatment, they share common psychological and behavioral symptoms as well as risk factors. It has been shown that experiencing negative emotions increases the risk of losing self-esteem and thus losing control over your eating behavior [11]. Excess body weight is an additional risk factor for eating disorders. Eating disorders are five times more common in overweight or obese adolescents than in normal-weight adolescents [12,13]. According to Stice et al., risk factors include the pursuit of the ideal, body dissatisfaction, dietary restrictions, impairment, and psychosocial factors [14]. Each of the factors listed above carries the possibility of developing at least one specific eating disorder in the future. Eating disorders have a significant negative impact on quality of life and can lead to metabolic, autoimmune, autoinflammatory and dermatological diseases due to the consequences of malnutrition and/or overeating [15,16,17]. However, the most important consequence of eating disorders is increased mortality, especially among patients with anorexia and bulimia [18,19].

The criteria for diagnosing eating disorders have changed over the years. One of the screening tools commonly used to detect eating disorders is the EAT-26 [20]. EAT-26 contains a three-factor structure. The “Diet” factor consists of 13 questions and is characterized by an analysis of calorie, carbohydrate and sugar content, which is motivated by the desire to be slimmer. The factor “Bulimia and preoccupation with food” consists of six items and is described by the tendency to purge after meals and excessive thinking about food, and seven items belong to the factor “Oral control”, which reflects the tendency to self-control eating. The last four questions refer to eating behaviors during the last 6 months [21].

The aim of this study was to determine the relationship between the impact of self-esteem and life satisfaction on the occurrence of eating disorders among adolescents in three areas, such as control, weight loss, and bulimia and food preoccupation.

## 2. Materials and Methods

In total, 233 primary and secondary school students aged 12 to 19 were surveyed. They all came from the Subcarpathian region. The average age of the respondents was 15.46 ± 2.33 years. Among the respondents, 37% were boys and 63% were girls. The mean body mass index (BMI) of the subjects was on average 20.84 ± 4 kg/m^2^, and the average BMI desired by the subjects was 19.72 ± 3 kg/m^2^. Each of the participants provided informed consent to participate in the study, and the consent of the legal guardian was also obtained. Lack of consent and an age under 12 or over 19 excluded participation in the research. The study was conducted according to the guidelines of the Declaration of Helsinki and approved by the Bioethics Committee at the University of Rzeszow (Resolution No. 2022/071, date 15 June 2022). The study was conducted in 2022, when restrictions related to the COVID-19 pandemic were lifted. Isolation had no effect on the test results.

To examine the risk of eating disorders, the Eating Attitudes Test-26 questionnaire was used. It explores the risk of eating disorders in three areas: dieting, bulimia and food preoccupation, and oral control. It is a standardized survey consisting of 26 questions relating to feelings and 4 questions related to behaviors occurring during the last 6 months in the examined person, such as losing more than 10 kg of body weight or a sense of loss of control over food consumption. This scale was developed in 1982 by David M. Garner for the early detection of eating disorders. The study used the Polish adaptation from 2016, developed by Rogoza et al. [22].

The Self-esteem Scale (SES) of M. Rosenberg consists of 10 statements that relate to your beliefs about yourself. It is a tool that allows you to assess levels of self-esteem. The respondent is asked to indicate to what extent he or she agrees with each of the statements [23]. This study used the Polish adaptation by Łaguna et al. [24].

The Cantril scale has the form of a ladder, with degrees from 0 to 10 defining the level of satisfaction, with 0 being the worst and 10 the best satisfaction. It is assumed that a score <6 is defined as dissatisfaction, and >6 means satisfaction with one’s life [25].

In addition, the respondents were asked to provide their height and current and desired body weight. On this basis, the current and desired Body Mass Index (BMI) was calculated, and their values were found on the WHO charts [26].

Statistical analysis of the collected data was performed in the Statistica 13.3 package. Only non-parametric tests were used for the analysis of variables. The choice of this type of test was conditioned by the failure to meet the basic assumptions of parametric tests, i.e., compliance of the distributions of the examined variables with the normal distribution, which were verified by the Shapiro–Wilk W test. For most numerical variables, descriptive statistics were calculated: mean, median, minimum, maximum, first and third quartiles, and standard deviation. The Mann–Whitney U test was used to assess the differences in the average level of a numerical trait in the two populations. The correlation of two variables that did not meet the criterion of normal distribution was determined using Spearman’s rank correlation coefficient. The level of statistical significance was *p* < 0.05.

## 3. Results

It was shown that men obtained higher results of satisfaction with their appearance (*p* < 0.001), body weight (*p* = 0.007) and life (*p* < 0.001) compared to women. Compared to men, women scored more on the SES scale (*p* = 0.001); therefore, they had lower self-esteem and a higher risk of eating disorders—EAT scale (*p* < 0.001). Regarding eating disorders, women also scored higher on dieting (*p* = 0.034), bulimia and food preoccupation (*p* = 0.005) and oral control (*p* = 0.001) (Table 1).

There was a statistically significant negative relationship between the age of the subjects and oral control EDs (*p* = 0.035; R = −0.14), and a positive relationship between the age of the subjects and bulimia and food preoccupation (*p* = 0.038, R = 0.14). The older the respondents were, the lower their score in the area of eating disorders—oral control. Conversely, older people scored higher in the category of bulimia and food preoccupation among ED areas (Table 2).

It was shown that people who lost 10 kg or more were on average younger (14.87 years old) than people who did not lose such an amount of body weight (15.64 years old). The described difference in the age of the subjects from the two groups was statistically significant (*p* = 0.039). The influence of age on losing more than 10 kg is shown in the Appendix A.

There were no differences in the ages of people treated and not treated for eating disorders (*p* = 0.052), although this result was close to the threshold of significance. People treated for this reason were on average slightly older (16.5 years old) than those not treated (15.37 years old).

People with lower self-esteem (more points on the SES scale) were less satisfied with their appearance and their body weight. It also showed that people with lower self-esteem were more likely to overeat in the last 6 months when they felt they could stop, vomit or exercise more than 60 min a day to lose weight or control weight. People with lower self-esteem had a higher risk of eating disorders—EAT-26 scale in all three areas (Table 3).

People who had lost 10 kg or more in the last 6 months had lower self-esteem (average 25.02 points on the SES scale) than people who had not lost weight (23.53 points). This difference was statistically significant (*p* = 0.010). The descriptive data related to weight loss of 10 kg or more and self-esteem are in Appendix A.

In addition, it was shown that people who were treated for eating disorders had lower self-esteem (average 25.9 points on the SES scale) than people who were not treated for this reason (23.68 points). This difference was statistically significant (*p* = 0.023). The correlation between treatment for eating disorders and self-esteem is shown in Appendix A.

The self-esteem of the surveyed people was differentiated by their satisfaction with their appearance (*p* < 0.001), satisfaction with their body weight (*p* < 0.001), risk of eating disorders—based on the EAT-26 scale in the area of slimming (*p* < 0.001), and controls (*p* = 0.017). The higher the respondent’s self-esteem, the higher their satisfaction with their appearance and body weight was. Risk of eating disorders—the EAT-26 scale was statistically significantly more frequent in people with low self-esteem than in people with average and high self-esteem. This was similar in the ED area—slimming. The degree of control, however, differed statistically significantly among people with low and high self-esteem (Table 4).

## 4. Discussion

In our study, there was a positive correlation between eating disorders and the female gender. Studies by other authors also confirm this relationship [27,28,29]. Erol et al. showed that men’s BMI was not related to their eating attitudes and behaviors [30]. Compared to men, however, women’s eating attitudes and behavior were strongly related to their BMI [31]. Girls experience more ED symptoms than boys [32]. Important conclusions reached by Holm et al. showed that although girls and boys aged 9–11 have similar levels of eating disorders, around the age of 14 the level of symptoms increases in females, while males remain stable [33]. In addition, studies have shown that as BMI increases, the risk of developing eating disorders and body dissatisfaction increases, especially in women [34]. The higher the BMI, the lower the body image satisfaction, which increases the risk of eating disorders. Studies also show a correlation between dissatisfaction with body image and the occurrence of binge eating, which was also confirmed by the results of our own study [35].

The age of diagnosis of eating disorders is decreasing. In a review by van Eeden et al., 31 publications were analyzed (systematic review and meta-analysis) from 1980 to 2019. It was found that over the years the incidence among persons aged under 15 years has increased. The reason may be the more effective diagnosis of the disease and taking into account younger and younger subjects [36]. A study conducted in primary care centers in the United Kingdom between 2003 and 2018 among 10- to 19-year-olds showed an increase in the incidence of EDs in the 13–16 age group. The statistics referred to both bulimia and anorexia. In this study, the authors were not sure whether the changes were related to the real lowering of the age of ED onset or were related to the increasing awareness and more frequent diagnosis of younger and younger groups [37]. Many studies suggest that the age at which eating disorders occur, especially in the area of bulimia, is decreasing. In a publication from 2022, meta-analytical epidemiological estimates of the age at the onset of mental disorders were made. The peak age at the onset of bulimia nervosa was 15.5 years. A similar result was obtained for anorexia nervosa, which was also 15.5 years old. For comparison, the peak age at onset of binge eating disorder was 19.5 years. In this study, the proportion of onset by 14 years, 18 years and 25 years for bulimia nervosa was, respectively, 16.0%, 45.3% and 82.9%. However, for anorexia nervosa, the figures were, respectively, 18.2%, 55.2% and 78.7% [38]. van Son et al. analyzed the results of a Dutch primary care study and compared the age at which women developed BN. In the years 1985–1989, it was the age range between 25–29, while in the years 1995–1999 it was 15–24 years [39]. Similar results were obtained in a study of 793 Italian BN patients in an outpatient eating disorder ward between 1985 and 2008. People born in the years 1970–1972 had an average age of onset of 18.5 years, while in people born in the years 1979–1981 it was 17.1 years [40]. In our study, weight loss above 10 kg concerned the youngest girls. This may confirm the trend that eating disorders are becoming more and more common in younger and younger people, especially in the area of bulimia. This is an important aspect for future risk factor research and prevention programmes.

It should be noted that people with abnormal eating habits are at high risk of eating disorders, which are characterized by a psychopathological attitude and accompanying groups of syndromes involving weight changes or physiological disorders, including anorexia nervosa, bulimia and atypical eating disorders [41]. For example, patients with eating disorders tend to overexert themselves in terms of exercise amount, frequency, or both [42]. Although physical activity is associated with health, in the case of eating disorders, it can carry greater losses [43]. In a study of 61 female triathletes, Elbourne et al. found a significant positive correlation between mandatory exercise and eating disorders [44]. People with eating disorders, trying to take care of their appearance and weight, increase physical activity to minimize the effect of overeating. At the same time, exercise becomes a chore, fueling eating disorders and lower self-esteem. Physical activity becomes a negative aspect of EDs [45]. As the results of the study indicate, compulsive exercise is associated with eating disorders such as bulimia and anorexia nervosa, which is consistent with the results of our own research, where the surveyed people who exercised for longer more often indicated slimming, bulimia and food preoccupation [46]. A study by Di Lodovico et al. showed a relationship between physical activity and a distorted image of one’s own body [47]. The subjects increased the amount of exercise despite the decrease in body weight. Our own research showed that women who chose to exercise more than 60 min a day to lose weight or control their weight had a higher risk of eating disorders in the area of dieting, bulimia and food preoccupation, and better control on the EAT-26 scale. According to Pritchard et al., women most often engage in mandatory exercise to improve mood, tone, and health, while men engage in mandatory exercise to reduce tension and for pleasure and physical attractiveness [48]. It is worth noting that previous studies have shown that people who exercise regularly have lower self-esteem and are more dissatisfied with their bodies [49].

To a large extent, self-esteem is related to satisfaction with the appearance. In particular, low self-esteem determines dissatisfaction with the appearance of the body, and vice versa, low self-esteem leads to dissatisfaction and a negative perception of the figure [50]. In our own research, people with lower self-esteem were less satisfied with their appearance and body weight. Negative body image is also strongly associated with low self-esteem [51], with the greatest dissatisfaction seen in the binge eating group [52]. Low self-esteem is a common problem among teenagers and is associated with psychiatric problems such as depression and anxiety. Dissatisfaction with body image contributes to incorrect eating behavior and possible restrictions [53]. People with low self-esteem want to achieve self-esteem by controlling food, weight or body shape, and according to Brockmeyer et al. achieving low weight can be defined as winning, which is also strongly associated with excessive control over eating [54]. However, contrary to the results of our own research on body dissatisfaction, the study did not notice any major differences between the female and male sexes, and the results of the study showed similar dissatisfaction occurring in both sexes [55]. According to the authors, this may be due to the smaller number of studies on the male group.

Satisfaction with life plays a significant role as a risk factor for eating disorders. Satisfaction with appearance and self-confidence affects this indirectly. Doornik et al. found that life satisfaction is a plastic factor that changes with the severity of anorexia symptoms. To obtain effective treatment, you also need to improve your commitment to specific domains of life [56]. Eating disorders strongly affect people’s perception of life. However, other studies by the same author confirmed the hypothesis that adolescents with AN show relatively low satisfaction with significant areas of life unrelated to AN [57]. In a study of 29 girls undergoing treatment, it was found that after the end of therapy there was a positive correlation between life satisfaction and the patient’s level of treatment. The life satisfaction chart may become a new approach to measuring life satisfaction, which has shown promising measurement and therapeutic properties [58]. This is a new and valuable direction for future research. In addition, an assessment of satisfaction with life together with an assessment of the risk of developing an eating disorder may help to predict the need for treatment more effectively.

## 5. Limitation

The present study has some limitations. Respondents could subjectively complete the questionnaires, which may change the results. Another limitation is the small number of men in the study. This study, however, provides some direction for future research. All people from primary and secondary schools in the Subcarpathian region who gave their consent were tested. Each respondent had different conditions of private life, knowledge about the state of physical health, and emotional approach. We believe that it would be interesting to study a group in a lower age range, with the exclusion criteria presented above. It is also worth diversifying the income levels and living environments of children’s carers in terms of the impact on the quality and quantity of nutrition. Considering further research on this topic, researchers should increase the study group and include adolescents from different regions of the country.

## 6. Conclusions

Gender plays a significant role in the risk of eating disorders. Women, compared to men, showed lower self-esteem and at the same time a higher risk of eating disorders in all three areas. Men showed greater satisfaction with their appearance, body weight and their lives, and they also had higher self-esteem.Episodes of significant weight loss (>10 kg) concerned a group of younger students. The younger the person, the greater the probability of eating disorders, especially in the area of bulimia and food preoccupation and control.Low self-esteem correlated positively with significant weight loss (>10 kg), more frequent uncontrollable binge eating and exercising more than 60 min a day to influence appearance. People with lower self-esteem were more likely to be less satisfied with their appearance and weight and were more likely to be treated for eating disorders.Among adolescents, satisfaction with appearance and weight affects life satisfaction, as measured with the Cantril ladder, which showed a positive relationship with the risk of eating disorders in all three areas. Subjects dissatisfied with their lives binged, feeling that they could not stop.

## Figures and Tables

**Table 1 nutrients-15-01603-t001:** Effects of gender on satisfaction with appearance, weight, life and risk of eating disorders.

	Women			Men			Z	*p*
	Average	Median	Standard Deviation	Average	Median	Standard Deviation		
Satisfaction with your appearance	5.88	6.00	2.36	7.08	7.00	2.15	−3.91	<0.001
Satisfaction with your body weight	5.38	5.00	2.72	6.40	6.50	2.72	−2.68	0.007
Satisfaction with your life	5.97	6.00	2.46	7.17	7,00	1.99	−3.66	<0.001
Points on a scale of self-esteem SES	24.44	25.00	4.00	22.88	23.00	3.51	3.18	0.001
Risk of eating disorders—EAT-26 scale including:	24.02	21.00	11.81	18.47	17.00	7.13	3.69	<0.001
Dieting	14.62	12.00	7.72	11.91	10.50	4.61	2.11	0.034
Bulimia and food preoccupation	4.37	4.00	3.39	3.19	3.00	2.52	2.78	0.005
Oral control	5.03	4.00	4.23	3.37	2.50	3.37	3.26	0.001

Z-Mann–Whitney U test result; red indicates significant values (*p* < 0.05).

**Table 2 nutrients-15-01603-t002:** Influence of age on respondents’ answers to selected questions.

	R	*p*
Binge eating feeling like you may not be able to stop	−0.03	0.616
Vomiting to affect your weight or shape	0.02	0.726
Using laxatives, weight loss supplements, or diuretics to control your weight or shape	0.06	0.325
Exercising more than 60 min a day to lose weight or control weight	−0.05	0.439
Satisfaction with your appearance	−0.01	0.919
Satisfaction with your body weight	−0.03	0.695
Satisfaction with your life	−0.09	0.167
Scores on the SES self-esteem scale	−0.04	0.595
Risk of eating disorders—EAT−26 scale including:	−0.03	0.627
Dieting	−0.04	0.565
Bulimia and food preoccupation	0.14	0.038
Oral control	−0.14	0.035

R—Spearman test result; red indicates significant values (*p* < 0.05).

**Table 3 nutrients-15-01603-t003:** The impact of self-esteem on the SES scale on the risk of eating disorders.

	R	*p*
Binge eating feeling like you may not be able to stop	0.16	0.014
Vomiting to affect your weight or shape	0.18	0.006
Using laxatives, weight loss supplements, or diuretics to control your weight or shape	0.12	0.060
Exercising more than 60 min a day to lose weight or control weight	0.15	0.022
Satisfaction with your appearance	−0.52	<0.001
Satisfaction with your body weight	−0.39	<0.001
Risk of eating disorders—EAT-26 scale including:	0.33	<0.001
Dieting	0.30	<0.001
Bulimia and food preoccupation	0.15	0.019
Oral control	0.22	0.001

R—Spearman test result; red indicates significant values (*p* < 0.05).

**Table 4 nutrients-15-01603-t004:** The influence of self-esteem on the selected questions.

	Low Score			Average Score			High Score			H	*p*
	Average	Median	Standard Deviation	Average	Median	Standard Deviation	Average	Median	Standard Deviation		
Binge eating feeling like you may not be able to stop	1.96	1.00	1.37	1.72	1.00	1.07	1.58	1.00	1.15	3.06	0.217
Vomiting to affect your weight or shape	1.20	1.00	0.69	1.07	1.00	0.49	1.16	1.00	0.90	4.88	0.087
Using laxatives, weight loss supplements, or diuretics to control your weight or shape	1.16	1.00	0.61	1.15	1.00	0.66	1.19	1.00	0.91	0.47	0.789
Exercising more than 60 min a day to lose weight or control weight	2.29	2.00	1.58	2.01	1.00	1.42	2.03	1.00	1.49	1.95	0.377
Satisfaction with your appearance	4.94 *	5.00	2.44	6.87 *	7.00	1.82	8.06 *	8.00	1.93	49.17	<0.001
Satisfaction with your body weight	4.65 *	5.00	2.74	6.09 *	6.00	2.52	7.52 *	8.00	2.43	28.10	<0.001
Risk of eating disorders—EAT-26 scale including:	26.01 *	23.00	12.29	20.10	18.00	8.87	17.94	16.00	8.60	19.23	<0.001
Dieting	16.36 *	14.00	8.02	12.29	11.00	5.62	11.10	10.00	5.13	18.54	<0.001
Bulimia and food preoccupation	4.26	4.00	3.27	3.82	3.00	2.85	3.45	3.00	3.79	3.68	0.159
Oral control	5.39 *	4.00	4.50	3.99	3.00	3.57	3.39 *	2.00	3.73	8.19	0.017

H—test value Anova Kruskala–Wallisa; red indicates significant values (*p* < 0.05), * statistically significant differences in the post-hoc test (multiple comparisons).

## Data Availability

The data presented in this study are available upon request from the corresponding author. The data are not publicly available, as they include sensitive clinical data.

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
