# Peer review of "Low Self-Esteem and Life Satisfaction as a Significant Risk Factor for Eating Disorders among Adolescents"

_nutrients, 2023, doi:10.3390/nu15071603_

Round 1

Reviewer 1 Report

This study evaluated the association between self-esteem and life satisfaction with eating disorders in adolescents. I have the following questions and suggestions. 

Line 25 - The introduction is long and can be shortened. 

Line 46 – OSFED – expand this acronym

Line 83 – when was this study carried out? Pre-pandemic or post-pandemic. As during the pandemic, many children gained weight and also had worsening psychological disorders, this context is very important and should be noted.

Line 86 – as 2/3 of the participants were girls, was this female dominance adjusted in regression analysis?

Line 89 – Please mention if institution review board approval was obtained (exempted from approval)

Line 92 – please explain what is “he”

Line 126 – Table 1 – “Medium” – is it “mean”?

Line 227 – The major limitation here is the recall bias which was not addressed.

Line 232 – Exclusion criteria. Other than lack of consent, were there any other criteria for the exclusion?

Reviewer 2 Report

The manuscript entitled “Low self-esteem and life satisfaction as a significant risk factor for eating disorders among adolescents” investigated the association between appearance and life satisfaction with eating disorder. They concluded that women are more susceptible with greater risk with eating disorder.

The study is well-structured and clearly presented. The research question is relevant and timely.

My only suggestion for this manuscript is to check the English and revise their language style as there are grammatical errors which has been noticed throughout the text.

Overall, this study is well designed and drafted.
